# Experimental Evaluation of an RSSI-Based Localization Algorithm on IoT End-Devices

**DOI:** 10.3390/s19183931

**Published:** 2019-09-12

**Authors:** Rosa Pita, Ramiro Utrilla, Roberto Rodriguez-Zurrunero, Alvaro Araujo

**Affiliations:** B105 Electronic Systems Lab, ETSI Telecomunicación, Universidad Politécnica de Madrid, Avenida Complutense 30, 28040 Madrid, Spain

**Keywords:** localization, RSSI, wireless sensor networks, IoT, FRORF

## Abstract

In recent years, wireless sensor networks (WSNs) have experienced a significant growth as a fundamental part of the Internet of Things (IoT). WSNs nodes constitute part of the end-devices present in the IoT, and in many cases location data of these devices is expected by IoT applications. For this reason, many localization algorithms for WSNs have been developed in the last years, although in most cases the results provided are obtained from simulations that do not consider the resource constraints of the end-devices. Therefore, in this work we present an experimental evaluation of a received signal strength indicator (RSSI)-based localization algorithm implemented on IoT end-devices, comparing its results with those obtained from simulations. We have implemented the fuzzy ring-overlapping range-free (FRORF) algorithm with some modifications to make its operation feasible on resource-constrained devices. Multiple tests have been carried out to obtain the localization accuracy data in three different scenarios, showing the difference between simulation and real results. While the overall behaviour is similar in simulations and in real tests, important differences can be observed attending to quantitative accuracy results. In addition, the execution time of the algorithm running in the nodes has been evaluated. It ranges from less than 10 ms to more than 300 ms depending on the fuzzification level, which demonstrates the importance of evaluating localization algorithms in real nodes to prevent the introduction of large overheads that may not be affordable by resource-constrained nodes.

## 1. Introduction

Nowadays wireless sensor networks have become a key technology used in diverse applications such as environmental and disaster area monitoring, security, inventory management, healthcare monitoring, etc. The resource-constrained sensors that compose these networks collect information about the environment that surrounds them and are interconnected with the rest of the nodes of the network, making this technology a fundamental part of the Internet of Things (IoT) [1].

The rise of IoT applications has caused the appearance of lots of interconnected devices that allow the compilation of large quantities of data using sensor nodes. However, for some applications the information obtained is only useful if the location of the sensor nodes is known [2]. Therefore, an efficient and low-cost localization mechanism is necessary.

Achieving precise location information in an efficient way—in terms of energy consumption, processing overload and memory overload—is an important issue for wireless sensor networks (WSNs) that has long been studied in the literature [3,4]. This research has led to three main localization schemes depending on the area of deployment [5]: wide area localization (WAL) schemes, local area localization (LAL) schemes and ad hoc localization (AHL) schemes. WAL schemes provide global localization data that can be calculated by the network or by the node to be located. This scheme is used to achieve high accuracy and fast localization in user applications such as navigation. The Global Positioning System (GPS) is the most popular, but direct GPS signals are difficult to receive in an indoor environment. The second one, the LAL scheme, is typically based on the features of the underlying physical layer, such as ultrasound, infrared, Bluetooth, etc. These schemes use parameters like the time of arrival (ToA) to provide a location. Finally, in the AHL schemes, the problem of localization is completely different. It consists of estimating the location of the nodes by collaborating with each other. These schemes are divided on range-free and range-dependent techniques.

Although GPS is widely used for mobile applications where localization is required, it is not feasible for indoor environments and implies a large energy consumption and cost [6]. This has led to the development of several AHL localization techniques for resource-constrained sensor devices. The ones based on a received signal strength indicator (RSSI) have also been long studied in the literature [7], but extensive experimental data of these techniques using real sensor devices are still lacking. This is an important issue since computational and communication overhead introduced by these algorithms could not be affordable by some sensor devices. RSSI-based techniques are very common in these sensor networks, since they do not require any additional hardware—most wireless transceivers provide RSSI data—or any synchronization in the network end-devices. However, most of the previous analyses done on these localization algorithms do not consider the impact of processing them in the resource-constrained end-devices. In general, those localization algorithms that require a large processing time highly increase the energy consumption of the nodes since the time available to enter in low power consumption modes is reduced. Therefore, it is important to evaluate not only the accuracy of the localization algorithms, but also its costs in terms of processing times for resource-constrained devices.

Some of the most used RSSI-based localization algorithms are trilateration [8] and ring overlapping based on comparison of RSSI (ROCRSSI) [9]. Some experimental works have been done using these algorithms [10], demonstrating that the accuracy achieved using the ROCRSSI techniques is better than the accuracy achieved using trilateration when more than three anchor nodes are used. In addition, the fuzzy ring-overlapping range-free (FRORF) algorithm introduces fuzzy logic to the ROCRSSI technique to achieve a better accuracy while increasing the algorithm complexity. However, an increased algorithm complexity could lead to large processing times that could have an important impact in low power devices that should be considered. Therefore, in this work, we have implemented the FRORF [11] localization method in IoT end-devices to evaluate its accuracy and the algorithm processing time for different levels of fuzzification. We have followed a distributed approach in which the nodes calculate their own positions, opposite to a centralized approach in which the location is calculated by an unconstrained high-end host. In this way, some modifications, that are detailed in this document, have been done to the original algorithm to fit sensor devices constraints, and multiple tests have been carried out in three different scenarios.

This paper is organized as follows. Section 2 presents the related work in localization for WSNs. Section 3 fully describes the algorithm proposed, while Section 4 presents the full description of the hardware and software resources employed to implement the FRORF as well as the methodology followed during the tests. Section 5 presents the results obtained in the different tests, which are discussed in Section 6. Finally, conclusions are offered in Section 7.

## 2. Related Work

As stated before, several localization techniques have proven to be a suitable solution to obtain location data of sensor nodes [12]. On the one hand, there are some range-free techniques, such as distance Vector-Hop (DV-Hop) based ones, that have proven good positioning accuracy when no distance indicators are available in the nodes [13]. On the other hand, there are also several range-dependent techniques that have long been studied, such as those based on RSSI measures [14], ToA measures [15], and angle of arrival (AoA) measures [16]. The first one has been highly studied for WSNs since it does not require any special hardware or any synchronization technique, as opposed to ToA techniques which require high resolution synchronized time measurements. Finally, the AoA techniques estimate the position by measuring first the angle of arrival of various signals from neighboring nodes. Nevertheless, it could result unsuitably for urban environments because the set of antennas can be blocked with a reflected signal that does not come from the transmitter node.

As a feasible solution for WSNs and IoT end-devices, many RSSI-based localization algorithms have been developed. Most of the existing works present the accuracy based on simulation results [17,18,19]. In addition, experimental data based on real deployments are also presented in some works. For example, localization performance of different wireless technologies has been evaluated [2]. In this case, experiments were performed using the trilateration method in indoor environments, with two scenarios: 10.8 m × 7.3 m and 5.6 m × 5.9 m. Other methods such as min-max method, maximum likelihood and statistical analysis have also been evaluated in experimental works for indoor and outdoor scenarios [20]. Marco Passafiume et al. [21] also presents an extensive experimental evaluation of a calibration-free localization algorithm, resulting in mean errors close to 1 m for indoor scenarios. Another extensive experimental work is presented by Xiaowei Luo et al. [22], in which different localization algorithms are evaluated for construction jobsite scenarios.

However, in these works it is not clear whether the localization algorithm is executed by the resource-constrained node to be located or by an external high-end unconstrained host. Executing the localization algorithm in the end-device is an important issue for IoT end-devices. A high computational complexity could result in unaffordable energy consumption leakages and also has an important impact in wireless communications performance [23].

For this work, we focus on the FRORF [11] algorithm since simulation results demonstrate a good accuracy and a significant improvement compared to ROCRSSI [9] and trilateration [8] localization techniques. As far as we know, no experimental works have been carried out for this algorithm running in a distributed way on resource-constrained end-devices, which is the main contribution presented in this paper.

## 3. Proposed Algorithm

In this section, the original FRORF algorithm—which is the one used as a baseline—is presented first. Following, we present the modifications made to this algorithm to adapt it to IoT resource-constrained devices.

### 3.1. Fuzzy Ring-Overlapping Range-Free Algorithm

The FRORF algorithm calculates the position of a node by delimiting the area where it is located. This algorithm uses two types of nodes: the fixed nodes that form the network infrastructure, also called anchor nodes, and the nodes that calculates their own position, which we call sensor nodes. In order to delimit the area of location a regionalization of the space is done through three steps:
Distance between anchor nodes: Each anchor node traces N-1 circles, N being the number of anchor nodes. The radius of those circles or rings are the distances between each pair of nodes and are numbered from 1 to N, with the interior ring being the first. The numbering of the interior area of a ring will be defined by its circle number as shown in Figure 1.Localization regions: The intersections between the different rings divide the space into the different localization regions. Each one of these regions is associated with a unique area number to differentiate them from each other.Area codes: This code is composed of the sequence of numbers of the rings that make up the region. Figure 2. Regions and area codes show two shaded regions, regions 1 and 2. The region 1 code is (3,3,1) which is the intersection of rings 3, 3 and 1 of nodes 1, 2 and 3 respectively. In the case of region 2, it is inside ring 2 of node 1 and inside the ring 1 of the nodes 2 and 3, so its code is (2,1,1). Finally, the regionalization map consists of the set of all the possible area codes.


To obtain the regionalization map we assume that the anchor nodes have self-known fixed positions so they can transmit them to the sensor node. Once the sensor node has obtained the positions from the anchor nodes it can create a regionalization map with the method presented above.

Then, when the regionalization map is determined, it is possible to obtain the area in which the sensor node is located. To delimit it, the algorithm uses the RSSI measurements taken by the sensor node from the packets transmitted by the anchor nodes. The measured RSSI values estimate the distances between the nodes, since the propagation losses increase with the distance. Therefore, it is possible to estimate a distance from the sensor node to each anchor node by a simple exchange of wireless packets, which allows a node to obtain the region in which it is situated. In order to obtain the code of the area of this region, the estimated distances to each anchor node and the radius of the regionalization rings are compared as follows:
(1)$$D_{n}\leq R_{ni}$$
where,
$D_{n}$ corresponds to the distance between the sensor node and the anchor node *n*;$R_{ni}$ corresponds to the radius of the ring *i* of the anchor node *n*.

In this way, the code of the area of the sensor node to be located consists of the set of the largest *i* rings that fit the comparison in Equation (1) for each anchor node *n*.

However, in real conditions the RSSI measurements are affected by multiple phenomena such as reflections, interferences, fading, etc. These phenomena deteriorate the signals causing the loss of monotonicity between the RSSI and the distance.

To compensate this effect, the FRORF algorithm incorporates fuzzy logic. So, it does not estimate the position of a node in a single region, but it estimates one or several degrees of belonging to the different regions, depending on the certainty that they give.

First, the area formed by two rings is defined as the intersection between LT = (0, β)—interior area of the exterior ring—and GT = (α,∞)—exterior area of the interior ring—as represented in Figure 3. Then, these intervals are defined as fuzzy sets: $\tilde{LT}=\{\left(x,\;\mu_{LT}\left(x\right)\right)|x\in R^{+})\}$ and $\tilde{GT}=\{\left(x,\;\mu_{GT}\left(x\right)\right)|x\in R^{+})\}$. Since $\mu_{LT}\left(x\right)$ and $\mu_{GT}\left(x\right)$ are the degree of belonging to an interval, Equations (2) and (3) are applied to determine it.
(2)$$\mu_{LT}\left(x\right)=\{\begin{array}{cc} 1 & if\;x\leq \beta \left(1-P\right)\\ \frac{\left(1+P\right)\beta -x}{2P\beta } & if\;\beta \left(1-P\right)<x<\beta \left(1+P\right)\\ 0 & if\;x\geq \beta \left(1+P\right) \end{array}$$
(3)$$\mu_{GT}\left(x\right)=\{\begin{array}{cc} 0 & if\;x\leq \alpha \left(1-P\right)\\ \frac{\left(1+P\right)\beta -x}{2P\beta } & if\;\alpha \left(1-P\right)<x<\alpha \left(1+P\right)\\ 1 & if\;x\geq \alpha \left(1+P\right) \end{array}$$
where,
β is a positive real-value corresponding to the radius of the exterior ring as shown in the Figure 3;α is a positive real-value corresponding to the radius of the interior ring as shown in the Figure 3;x is a positive real-value corresponding to the estimated distance between the node to be located and the anchor node as shown in the Figure 3;P is the level of fuzzification, a value between 0 and 1 that controls the width of the fuzzy region in the vicinity of ring boundaries, β and α.

The fuzzy ring is defined as shown in Equation (4) and the equation associated to the belonging function is presented in Equation (5).
(4)$$RI=LT\cap GT=\{\left(x,\mu_{RI}\left(X\right)\right)|X\in R^{+}\}$$
(5)$$\mu_{RI}\left(x\right)=\mu_{GT}\left(x\right)+\mu_{LT}\left(x\right)-1$$


Then, the fuzzy-ring set of an anchor node B_i_—a set that contains all the fuzzy rings of B_i_—is defined as shown in Equation (6).
(6)$${\tilde{RS}}_{I}=\left\{\left(j,{\mu_{RI}}_{j}^{i}\right)|{\mu_{RI}}_{j}^{i}>0,\;j\in \left\{0,\ldots ,N\right\}\right\}$$


The next step consists of determining the degree of belonging to a region. It is estimated as the cartesian product of the regions sets of the rings that compose it as shown in Equation (7).
(7)$$\tilde{RM}={\tilde{RS}}_{1}\;X\;{\tilde{RS}}_{2}X\ldots X\;{\tilde{RS}}_{n}=\left\{\left(c,\mu_{RM}\left(c\right)\right)\right\}$$
where,
c is the code of the region;$\mu_{RM}\left(c\right)$ is the degree of belonging to the region with code c defined as $\mu_{RM}\left(c\right)=\prod_{k=1}^{N}{\mu_{RI}}^{k,j_{k}}$.

Finally, the algorithm estimates the absolute position of the node from the centers of gravity (CoG) or centroids of the regions weighted according to their degree of belonging, thus smoothing the errors. A centroid is defined as the geometric center of an object.

### 3.2. FRORF Implementation

In this work, we have implemented the FRORF algorithm to evaluate its performance in a simulation environment before implementing it in IoT end-devices with limited resources. In addition, due to the complexity of this algorithm, a series of modifications have been made to fit it on a resource-constrained device. The algorithm implementation for the simulations and for the end-devices as well as the modifications done are explained below.

• FRORF simulations.

In order to simulate the FRORF algorithm, we use the grid scan method [24] to determine the CoGs of the regions. This method approximates the area of the regions by dividing the location space into a uniform grid of cells where the center of each cell, called the cell point(s), represents the total area of it.

A regionalization map D is formed by a set of localization regions $C_{k}$—the intersection of anchor rings—which are in turn formed by a set of cells. A cell is considered to belong to a given ring if the distance from the cell point s with coordinates ($x_{s}$, $y_{s}$) to the center of the ring is less than or equal to its radius. In this way, the grid of cells is scanned point by point to identify the localization regions they belong to. By using this method, the CoG coordinates ($x_{C_{k}}$, $y_{C_{k}}$) for each region $C_{k}$ are obtained from the equations below:
(8)$$x_{C_{k}}=\frac{\sum_{s}x_{s}}{N_{C_{k}}}\forall \;s\in C_{k},$$
(9)$$y_{C_{k}}=\frac{\sum_{s}y_{s}}{N_{C_{k}}}\forall \;s\in C_{k},$$
where, $N_{C_{k}}$ is the number of cells that belongs to each $C_{k}$ region. As an example, Figure 4 presents two regions obtained using this method (Figure 4b) compared to the ideal regions (Figure 4a). In addition, we present in Figure 5 the CoG of the different regions of two different maps which are represented by small red circles. As shown, the CoG is not necessarily located inside the region to which it belongs; in those regions that are very curved, the CoG is tending to the center of the circumference.

It should be noted that the greater the granularity used in the grid scan method—more and smaller cells—the better the CoG approximation is, which could have an important impact in localization accuracy. However, the computational cost also increases because there are more cells to process.

On the other hand, in the simulations the degree of irregularity (DOI) [9] model has been adopted to calculate the signal power received by the sensor nodes at any point in the coverage range of the anchor nodes. It is concretely done from the following expression:
(10)$$\mathrm{RSSI}=P_{TX}{\left(\frac{\lambda }{4\pi d}\right)}^{2}K\left(\theta \right)$$
where,
RSSI [mW] is the received signal power;$P_{TX}$ [mW] is the transmitted signal power;λ [m] is the wavelength of the signal in meters;d [m] is the distance between the receiver and the transmitter in meters;K(θ) is the coefficient that represents the propagation loss as a function of the direction.

The K(θ) coefficient is calculated using the following expression:
(11)$$K\left(\theta \right)=\{\begin{array}{cc} 1 & \theta =0\\ K\left(\theta -1\right)+\left(rand\;x\;\mathrm{DOI}\right) & \theta \;is\;a\;positive\;integer \end{array}$$
where θ∈[0,360] and rand is a random number uniformly distributed in the interval [−1, 1]. The DOI parameter is used to control the signal propagation irregularity and is defined as the maximum variation of the signal strength for a displacement of one degree in the direction of propagation. Thus, as the DOI increases, the radio propagation pattern becomes more and more irregular. As can be seen in Figure 6 where the RSSI is shown in dB as a function of θ.

It should be noted that simulating a network generates a different propagation pattern for each node—anchors and sensor node—but without any correlation between them (i.e., if the sensor node is at a point on the line that joins two anchors, B_1_ and B_2_, it is possible that a message for B_1_ suffers an extra attenuation, while one of B_2_ with the same receiver has no extra attenuation, even though both come from the same direction). In a real situation, generally the effect of the attenuation on the message will also affect B_2_ to a greater or lesser extent.

• FRORF in IoT end-devices.

In this case, in order to determine the CoG a different approach has been implemented instead of the grid scan method. As stated before, the grid scan method implies a large computational cost since it needs to process all the cells of a region to obtain its CoG. For this reason, we have approximated the regions to single rectangles to reduce the computational load on the sensor nodes.

For doing this, the anchor rings are approximated to squares to obtain the region code. The sides of these squares are the diameters of the original anchor rings, while the centers are preserved as shown in Figure 7. Each square is defined as the intersection of four straight lines that we call $x_{h}$, $x_{l}$, $y_{h}$ and $y_{l}$, as shown in Figure 8.

Therefore, the localization regions are formed by the sum of the areas of all the rectangles resulting from the intersections between the different squares. For example, Figure 9 shows the approximation of the region (2,3,2). This region is the sum of the three rectangles shaded in yellow, obtained through the intersection of the rings—or squares in this case—2, 3 and 2 of nodes 1, 2 and 3, respectively.

To perform this approach, firstly, the sensor node calculates the area code in which it is located—obtained from anchor nodes positions and RSSI measures—as presented in Section 3.1. Then, an initial square region is defined by the sides $x_{1}$, $x_{2}$, $y_{1}$ and $y_{2}$, which are the most restrictive lines of the *i* squares that represent the area code, as presented below:
(12)$$x_{1}=\max_{i}\;x_{l_{i}},$$
(13)$$x_{2}=\min_{i}\;x_{h_{i}},$$
(14)$$y_{1}=\max_{i}\;y_{l_{i}},$$
(15)$$y_{2}=\min_{i}\;y_{h_{i}},$$


Figure 10 shows an example of these limits for the area code (2,1,1) in a three-node infrastructure. In this figure, the rings that represent the area code are in orange along with the square’s straight lines and the limited sides are surrounded by a grey circle. As a result, the initial region will be as presented in Figure 11.

Once the initial region is obtained, the inner squares are considered, which are defined by the previous numbers of the area code. For example, for a sensor node with the area code (2,1,1) the previous squares area code is (1,0,0); actually, the squares of value 0 do not exist and are not considered. Then, all the intersections of the initial region with the inner squares are calculated and the resulting area is subtracted from the initial region.

Figure 12 shows the final region of the previous example after subtracting the intersection of the inner squares—only one exists in this case—with the initial region. The orange square corresponds to the inner square 1 of node 1 and the shaded area is the final region.

Finally, the CoG of a rectangle with coordinates ($x_{R},\;y_{R}$) coincides with half of the value of its superior limits, as presented below:
(16)$$x_{R}=\frac{x_{h}}{2},$$
(17)$$y_{R}=\frac{y_{h}}{2},$$


Therefore, to calculate approximated CoG for a rectangle-based region, we obtain the arithmetic mean of the CoG of all the rectangles that form the region.

In addition, we realized that the FRORF algorithm is sometimes unable to calculate a position. There are two reason for this:
The estimated distance between the node to be located and the anchor nodes is larger than the radius of all exterior rings.The area code obtained does not correspond to any region, because the rings that compose it do not intersect.

Therefore, to solve this problem we decided to modify the algorithm to always estimate a position. For this reason, when this situation occurs, the anchor node with the lowest RSSI, that is, the theoretically furthest, is ignored and the position is recalculated without considering this anchor node. It should be noted that if only a single anchor node is available to estimate the position, the position of this anchor node is provided.

## 4. Materials and Methods

### 4.1. Hardware and Software

The nodes used to perform the experiments were the YetiMotes IoT devices [25]. These nodes are based on a high-performance low-power STM32L476RE microcontroller running at 48 MHz and equipped with 512 KB of flash memory and 128 KB of RAM memory. This microcontroller also supports low-power modes and peripheral interfaces such as UART, SPI, I2C, USB and a microSD card slot. The node has 2 accelerometers, a temperature sensor and 3 radio interfaces for 433 MHz, 868 MHz and 2.45 GHz bands. In this work, all the tests have been performed using the 433 MHz band.

The FRORF has been implemented in the YetiMotes using YetiOS [26]. This operating system (OS) is built on top of FreeRTOS and provides several features such as standard input-output (STDIO), a shell (YetiShell) and general-purpose input/outputs (GPIOs). The OS has advanced memory management, advanced process management, provides Linux-like device drivers and time management modules. YetiOS also provides a radio communication stack which is used to obtain the RSSI values from the anchor nodes.

As mentioned previously, the location of the nodes is determined by themselves, so each one must calculate its own position. Therefore, a distributed approach has been chosen instead of a centralized approach, where the location is calculated by an unconstrained high-end host, in order to evaluate the accuracy and processing times in IoT end-devices.

### 4.2. Methodology

#### 4.2.1. Distance Calculations

This section aims to explain in more detail how the distance between nodes has been obtained using the RSSI data. This estimation has been made with different procedures for the case of the simulator and for the implementation in the end-devices.

• FRORF simulations.

In the case of the simulator, the log-distance propagation loss model has been used, modelled by Equation (18).
(18)$$PL=P_{TxdBm}-P_{RxdBm}=PL_{0}+10\gamma \;log_{10}\frac{d}{d_{0}}+X_{g}$$
where,
*PL* corresponds to the total propagation loss measured in decibels (dB);*P_TxdBm_* corresponds to the transmitted power in dBm;*P_RxdBm_* corresponds to the received power in dBm;*PL*_0_ corresponds to the total propagation loss measured in a reference distance *d*_0_;*d* [m] corresponds to the distance between transmitter and receiver;*d*_0_ [m] corresponds to the reference distance used to measure *PL*_0_;γ corresponds to the propagation loss exponent;$X_{g}$ corresponds to the parameter that reflects the attenuation caused by the plane fade. In this case it is assumed to equal to zero.

Therefore, the following expressions have been used to obtain the distance of a node:
(19)$$\mathrm{RSSI}\text{\_}aa_{i,k}=P_{TxdBm}-PL_{0}-10\gamma \;log_{10}\frac{d\text{\_}aa_{i,k}}{d_{0}}$$
where,
$\mathrm{RSSI}\text{\_}aa_{i,k}$ corresponds to the signal power in dBm received by anchor node B_i_ from another anchor node B_k_;$d\text{\_}aa_{i,k}\;\left[m\right]$ corresponds to the distance between B_i_ and B_k._
(20)$$\mathrm{RSSI}\text{\_}as_{i,s}=P_{TxdBm}-PL_{0}-10\gamma \;log_{10}\frac{d\text{\_}as_{i,s}}{d_{0}}$$
where,
$\mathrm{RSSI}\text{\_}as_{i,s}$ corresponds to the signal power in dBm received by an anchor node B_i_ from the node to be located S;$d\text{\_}as_{i,s}\;\left[m\right]$ corresponds to the distance between B_i_ and S.

Establishing a relationship between the expressions, the following equation is obtained:
(21)$$\mathrm{RSSI}\text{\_}aa_{i,k}-\mathrm{RSSI}\text{\_}as_{i,s}=10\gamma \;log_{10}\frac{d\text{\_}aa_{i,k}}{d\text{\_}as_{i,s}}$$


Equivalent in natural units:
(22)$$\frac{\mathrm{RSSI}\text{\_}aa_{i,k}}{\mathrm{RSSI}\text{\_}as_{i,s}}={\left(\frac{d\text{\_}aa_{i,k}}{d\text{\_}as_{i,s}}\right)}^{\gamma }$$


From Equation (23), the algorithm obtains an estimate ${\mathrm{d\_est}}_{i,s}^{\left(k\right)}$ of the distance between the anchor node B_i_ and the node to be located S, using as reference the RSSI measurement of a message sent by another anchor node B_k_.
(23)$$d\text{\_}est_{i,s}^{\left(k\right)}=d\text{\_}as_{i,s}=d\text{\_}aa_{i,k}\sqrt[{\gamma }]{\frac{\mathrm{RSSI}\text{\_}aa_{i,k}}{\mathrm{RSSI}\text{\_}as_{i,s}}}$$
FRORF implementation.


In the case of the implementation in resource-constrained devices, the distance between nodes is obtained by a linear approximation, estimating a straight line on an experimental basis. It was decided to use this method instead of the previous one to simplify the calculations done by the end-devices and the wireless communication packets required between the nodes, thus reducing the computational cost. The experiment consisted of obtaining the average of 100 RSSI measurements per meter in a total of 15 m.

Finally, the data obtained are represented in order to make an approximate linear relation between the RSSI and the distance. The experiment was done for two different nodes obtaining the results shown in Figure 13 and Figure 14.

Therefore, the estimated distance is given by Equation (24) where the slope and offset are estimated as the average of the results obtained in the tests performed.
(24)$$d_{est}=-0.45\mathrm{RSSI}-17.45$$


However, a calibration process has been implemented that modifies the offset value. This consists of obtaining RSSI data in an equidistant zone to all the nodes to calculate the new offset as shown in Equation (25).
(25)Offset=distance_to_nodes−(0.45RSSI)


In addition, in an equidistant point the RSSI should theoretically be the same for all nodes. However, this does not happen. For this reason, the RSSIs have been adjusted to the maximum received one, adding to them the difference between the maximum value and it.

#### 4.2.2. Localization Algorithm Procedure

The procedure to obtain a position by the sensor node is explained as follows. First, it is necessary to exchange packets between the anchor nodes and the sensor node to obtain both the RSSI data and the position from the anchor nodes. The procedure performed by each type of node is detailed below:
Sensor node: It sends a data request broadcast packet for all the anchor nodes in range and waits for their responses. When a fixed time of 40 ms is reached, the sensor node should have received the response packet of the anchor nodes containing their position as well as the RSSI values. Then, based on the received data, the sensor node executes the full localization algorithm explained before to obtain a position. We have assumed that the sensor node has no previous data of the anchor nodes’ positions, so the localization algorithm must be fully executed each time. In this way, we have selected a flexible approach (i.e., the anchor nodes may be mobile, or the sensor node may be also mobile in an unknown anchor nodes’ infrastructure), that should provide the worst-case results in terms of processing load.Anchor nodes: These nodes are always listening. When a data request broadcast packet is received, they send a reply with their position data, the address and the RSSI of the received packet.

Figure 15 shows a scheme of the communications scheme previously explained. In addition, it is important to note that in order to avoid packet collisions, a different fixed delay has been included before sending a response in each anchor node.

#### 4.2.3. Experimental Scenarios

Once we have detailed the hardware and software resources and the communication scheme, we explain the methodology followed in our tests. The experiments were performed in a free-space outdoor environment. Three different scenarios with four anchor nodes were implemented: in two of them the dimensions were 9 m × 15 m and in the other one they were 15 m × 30 m. In the first case, the nodes were placed forming a rectangle (scenario 1) and a rhombus (scenario 3), and in the second case only a rectangle (scenario 2), as shown in Figure 16. In addition, in the three scenarios the nodes have line of sight (LoS) between them.

These scenarios were implemented in the simulator as well as in a real outdoor set-up for further comparison.

At the real setup, measurements were taken every meter on the *x*-axis for the three scenarios and every meter on the *y*-axis for scenarios 1 and 3. However, for scenario 2 measurements were taken every two meters on the *y*-axis. The obtained position was the average of 10 position measurements for scenario 1 and for the other two the average of 100 positions (Table 1).

In addition, nine different position values were obtained for each measure (level of fuzzification) for *p*-values ranging from 0 to 0.9, being the case *p* = 0 the equivalent of using ROCRSSI. It should also be noted that these data were given in geographical coordinates (latitude and longitude).

On the other hand, the three scenarios were implemented in the simulator with a DOI of 0.1, a grid of 20 × 20 and a *p*-value = 0.5, where the values were extracted from preliminary experiments with the aim of having the DOI as realistic as possible and the *p*-value to ensure the optimal performance of the algorithm.

## 5. Results

In this section, we present the results obtained from the tests performed in this work. Firstly, a comparison of the accuracy obtained for each of the scenarios in the outdoor set-up has been made according to the level of fuzzification. Figure 17 shows the mean error for different *p*-values for the three implemented scenarios. On the *x*-axis we represent the *p*-value that have been studied, while the *y*-axis represents the error in meters between the real position and the obtained position.

It is noticeable that for scenario 2, in which the anchor nodes are the most separated, the error obtained is considerably larger. On the other hand, in scenario 3, the error values present small variations with the *p*-value, while for the other two scenarios there is a larger error variation. However, in all cases the best accuracy is obtained for *p*-values close to 0.5.

In addition, the impact of the DOI in the average error for the three scenarios has also been evaluated. Figure 18 shows the increase in this error as a percentage, using as the reference the average error value for a DOI of 0 and *a p*-value = 0.5. As it can be seen, the error is highly increased with the DOI so it must be considered when evaluating the simulation results.

Figure 19, Figure 20 and Figure 21 show the error in meters of the position measured for the three scenarios for a *p*-value equal to 0.5. We have decided to use this *p*-value since, as presented in Figure 17, it achieves the lowest average error in the experiments we have performed.

These figures show the results obtained both from simulation and from the real outdoor set-up. In both cases, a map is presented showing the error with a scale of colors, where blue represents the smallest error and yellow the largest. Furthermore, in the case of the simulator, the *x* and *y* axes are represented in meters, as this is not a real scenario. However, in the case of the outdoor environment they are shown in geographical coordinates since these tests were performed in a specific location, and, therefore, the algorithm implemented in the nodes provides the positions in latitude and longitude values.

Additionally, as presented in Section 3.2, we used a rectangle-based regionalization approach instead of the grid scan method to obtain the CoG in the sensor nodes. For this reason, we have first evaluated the impact of using squares in the simulator, as it is important to know the impact of this approach in location accuracy in order to validate the results in the end-devices. We have redefined the way in which area codes are obtained in the simulator and squared regions are considered instead of circular ones. So, instead of using Equation (1) to determine the area codes, we use the conditions presented below in this case:
(26)$$x_{l}\leq x_{s}\leq x_{h}$$
(27)$$y_{l}\leq y_{s}\leq y_{h}$$


Table 2 shows a summary of the average error obtained in each of the scenarios, both for the simulator and for the outdoor set-up. It should be noted that the simulator calculates the error for the external areas of the infrastructure, so in comparing the average error of the simulator and the outdoor set-up only the interior area is considered.

In addition, as it can be seen, scenario 1 provides the most accurate position values and scenario 2 is the most inaccurate. Another thing to emphasize is that for the three scenarios the accuracy increases in the central areas of the network.

Finally, we have obtained performance data of the FRORF algorithm executing in the node. In Figure 22 the execution time of the algorithm is shown depending on the different *p*-values. Increasing the level of fuzzification has a large impact in the execution time of the algorithm. For example, for a *p*-value = 0.5 the execution time exceeds 300 ms which could have an important effect in the power consumption and in the performance of other tasks that could be running in the node.

## 6. Discussion

In this work we presented the accuracy results of the experiments carried out in simulations and in the implementation of the FRORF algorithm in resource-constrained devices, so it possible to observe differences between them. First, it was observed that better results are obtained for the simulator in terms of accuracy, which can be due to two main causes:
Methodology for calculating distance: The simulator uses a more complex method for calculating the distance between nodes, as presented in Section 4.2.1, and the more accurate these distances are, the better the position obtained.Methodology to estimate the RSSI: The simulator estimates the RSSI values using the DOI model, where the RSSI variations follow a uniform random distribution. Nevertheless, that is not the case, since there are many propagation phenomena that are very difficult to estimate and simulate.


However, it should be noted that in terms of overall behavior of the algorithm, the simulator provides similar results to the ones of implemented in IoT devices. The best and worst accuracies are obtained for the same scenarios. In addition, we have verified in the simulations that our proposed approach based on squared regions only increases 0.2 m the average error. In this way, our proposed CoG calculation method could be an interesting alternative to the grid scan method for resource-constrained devices.

On the other hand, different behaviors have been observed depending on the chosen topology. First, for the topology with the most separated nodes, the accuracy obtained is the worst. That may be caused due to the low RSSI linearity for large distances. In addition, where the regions are large there is a high probability of getting several positions in the same single region. The fuzzification tries to adjust this with some degrees of probability, but it has some limitations. On the other hand, comparing scenarios 1 and 3, both with the same dimensions, a better precision is obtained for scenario 1. This could happen because in the case of scenario 3 the number of regions in the interior zone of the network is lower than in the case of scenario 1, which means that there are more possible positions in each region.

It is also interesting to evaluate the behavior for different *p*-values. In all cases, the best accuracy is obtained for values close to 0.5. This is because fuzzy logic corrects well for relatively small variations, but in the case of 0.9, for example, it almost doubles the distances that form the fuzzy ring, which distorts the regions too much.

So, according to the experimental results the simulator gives us a good first approach of the algorithm behavior. However, it is very difficult to accurately simulate the RSSI values needed by localization algorithms. For this reason, it is important for evaluation of future RSSI-based localization algorithms to perform the tests in real scenarios. In our case, the FRORF algorithm has been implemented, obtaining different accuracy results to those simulated. To improve this accuracy, more complex methods could be used in the calculation of distance, or the combination of this technique with others.

Finally, the execution time of the algorithm running in the nodes has been evaluated. For low values of the parameter the execution time is much lower than for high values. This is because the higher the *p*-value, the more regions are considered to estimate a position, therefore increasing the amount of information to be processed. In addition, the original version of the algorithm is more complex than the implemented one, therefore without modifications the execution time would be higher. For example, in the case of *p* = 0.5 the most accurate results are obtained but at the expense of a very high execution time, close to 320 ms, which could introduce a large processing overhead in the IoT end-devices. So, it would be necessary to evaluate if it is worth it to use complex algorithms in resource-constrained devices, due to their high consumption.

## 7. Conclusions

In this work, we implemented and evaluated a FRORF-based localization algorithm in a real outdoor deployment. This algorithm was modified in order to fit IoT end-device constraints, with low memory and computing capabilities. Through experimentation in three different scenarios, we have evaluated the localization accuracy obtained with this algorithm, comparing simulation results for different DOI values and end-device implementation results. The effects of the fuzzification parameter *p*-value has also been evaluated, obtaining the most accurate results for values close to 0.5 and for the smaller topologies. We have also realized that the average errors obtained present large differences depending on the scenario, but they are not significantly affected by our CoG calculation approach for resource-constrained devices based on squares.

The differences between the results obtained by simulation and those obtained in a real outdoor set-up demonstrate the importance of evaluating RSSI-based localization algorithms in real scenarios. In addition, we evaluated the execution times of the implemented algorithm running in the nodes. These times vary from less than 10 ms to more than 300 ms depending on the value of the fuzzification parameter. This demonstrates that the actual implementation of the localization algorithms for WSNs is therefore a key element that should be considered for resource constraint nodes. In the case presented in this paper, a fuzzification value of *p* = 0.5 provides the best results in terms of accuracy, although it also results in large execution times, over 300 ms. Therefore, it is important to continue conducting experimental studies that provide enough information to developers to be able to choose in each case the most appropriate localization algorithm for a specific scenario and a specific hardware platform.

## Figures and Tables

**Figure 1 sensors-19-03931-f001:**
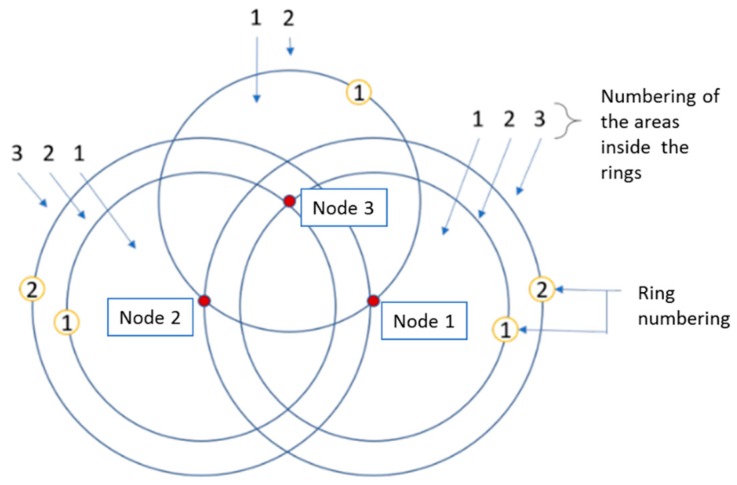
Ring numbering.

**Figure 2 sensors-19-03931-f002:**
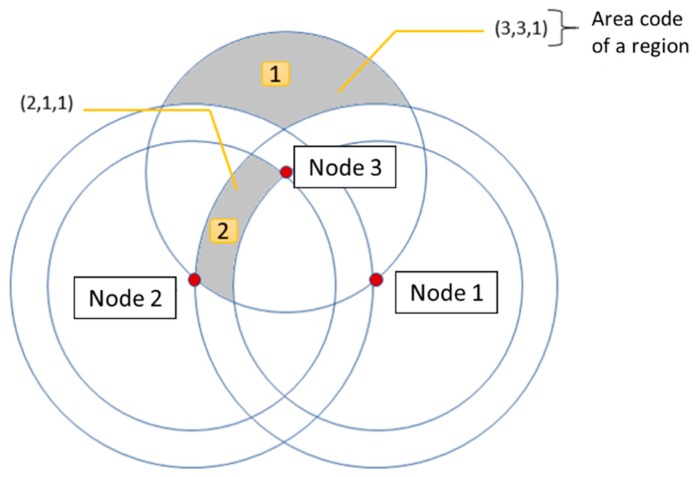
Regions and area codes.

**Figure 3 sensors-19-03931-f003:**
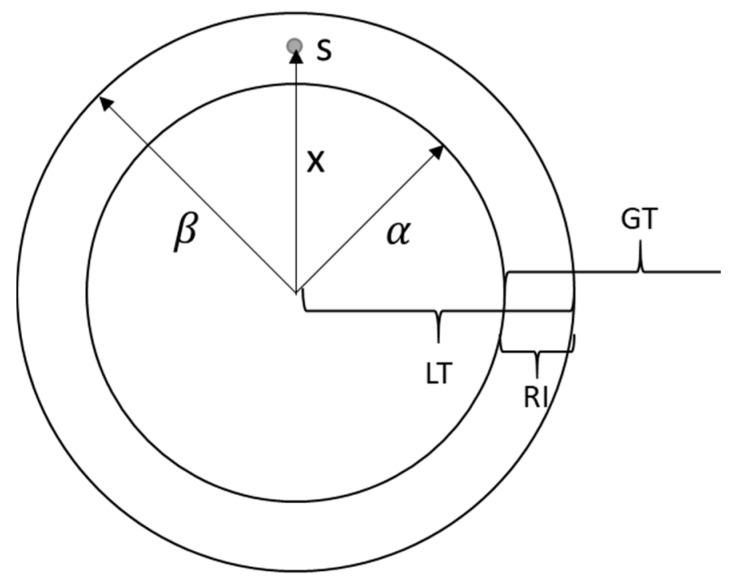
Parameters of a ring.

**Figure 4 sensors-19-03931-f004:**
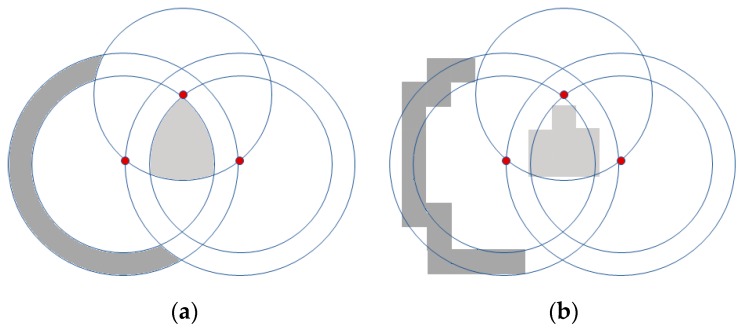
Example of the regionalization using the grid scan method. (**a**) Ideal region; (**b**) method grid scan with low granularity.

**Figure 5 sensors-19-03931-f005:**
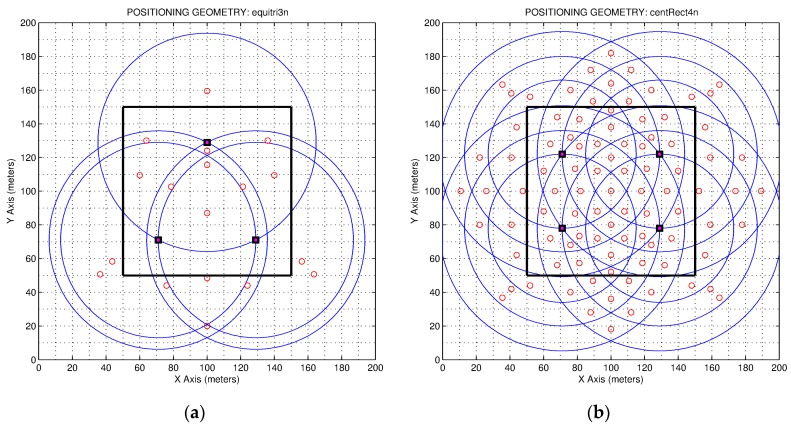
Centers of gravity (CoG) of the regions of two regionalization maps. (**a**) Triangular geometry of anchor positioning; (**b**) rectangular geometry of anchor positioning.

**Figure 6 sensors-19-03931-f006:**
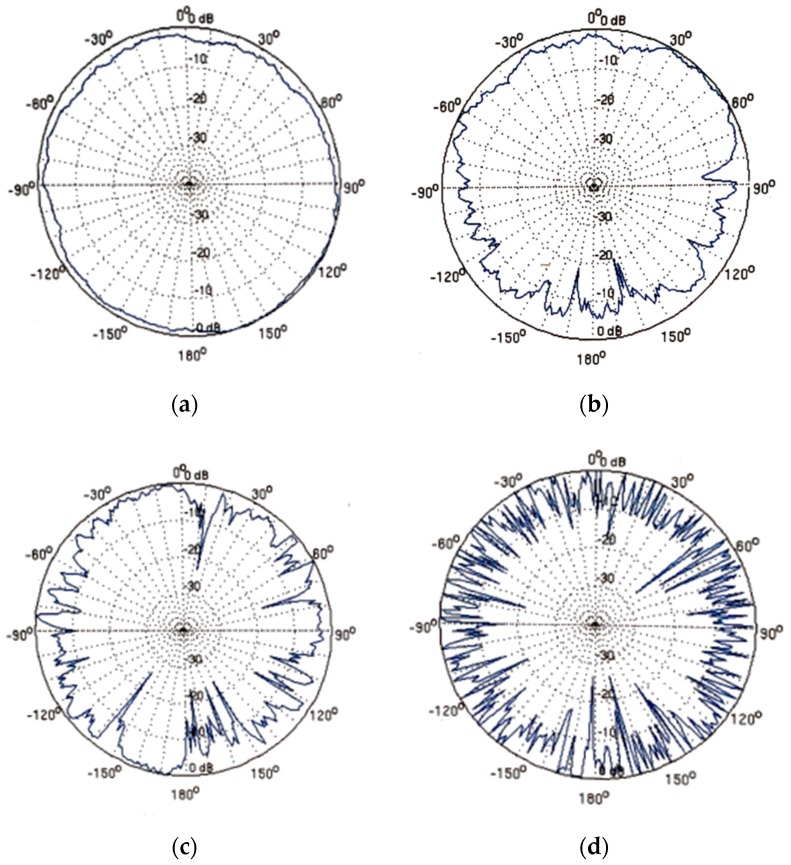
Radio propagation patterns for different degrees of irregularity (DOI). (**a**) DOI = 0.05; (**b**) DOI = 0.1; (**c**) DOI = 0.2; (**d**) DOI = 1.

**Figure 7 sensors-19-03931-f007:**
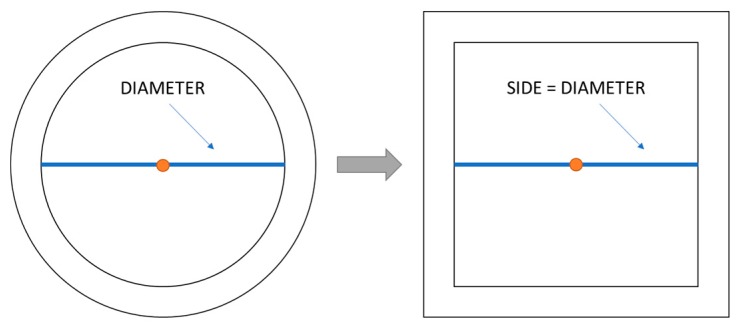
Approximation of the anchor rings by squares.

**Figure 8 sensors-19-03931-f008:**
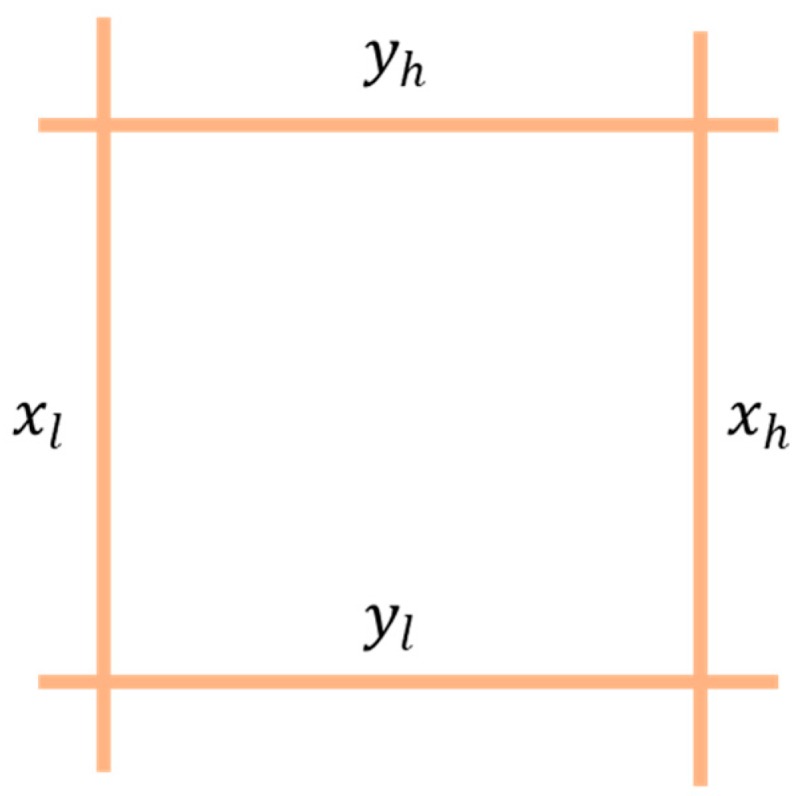
Square defined as the intersection of four straight lines.

**Figure 9 sensors-19-03931-f009:**
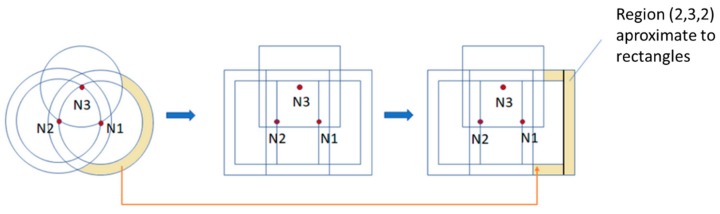
Approximation of a region by rectangles.

**Figure 10 sensors-19-03931-f010:**
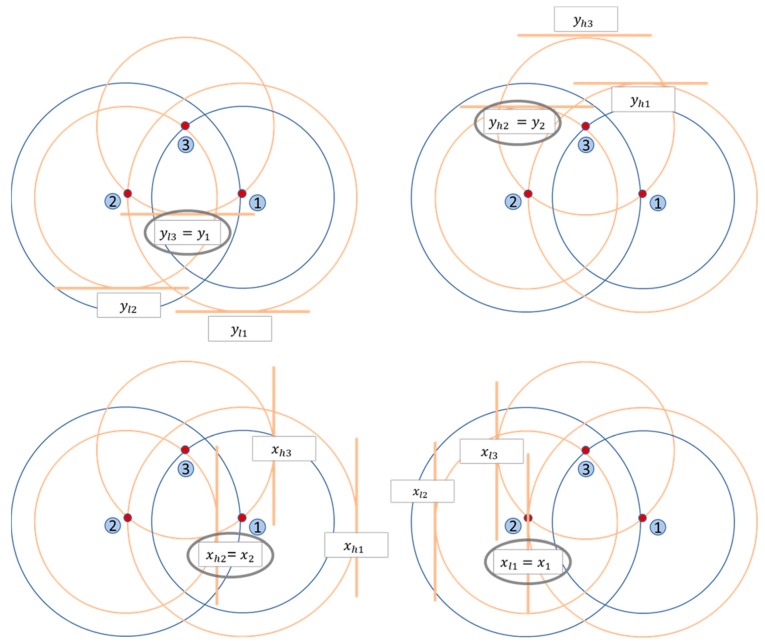
Restrictive limits of the area (2,1,1).

**Figure 11 sensors-19-03931-f011:**
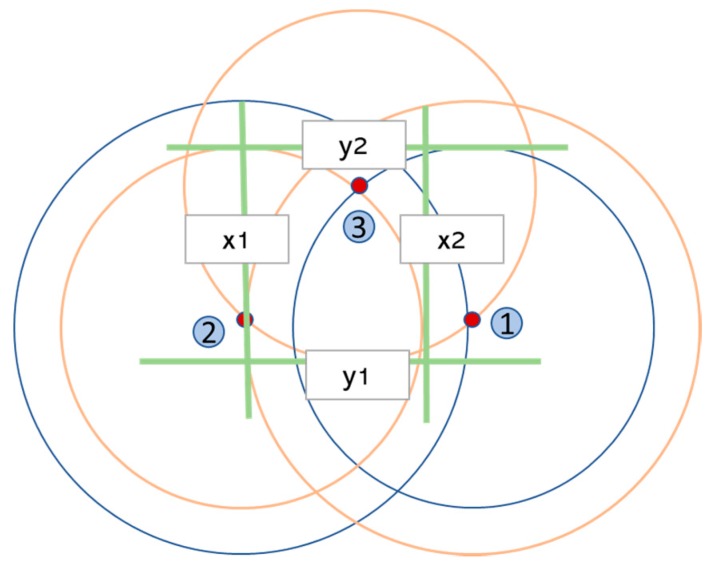
Initial region of the area (2,1,1).

**Figure 12 sensors-19-03931-f012:**
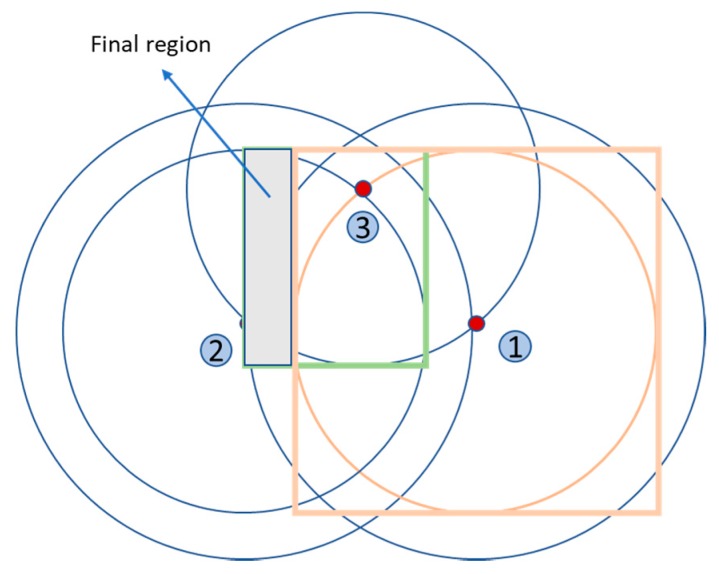
Final region of the area (2,1,1).

**Figure 13 sensors-19-03931-f013:**
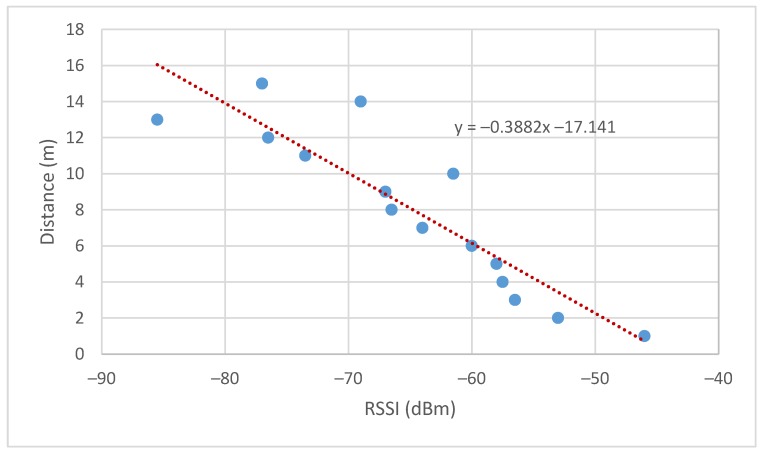
Received signal strength indicator (RSSI) vs. distance node 1.

**Figure 14 sensors-19-03931-f014:**
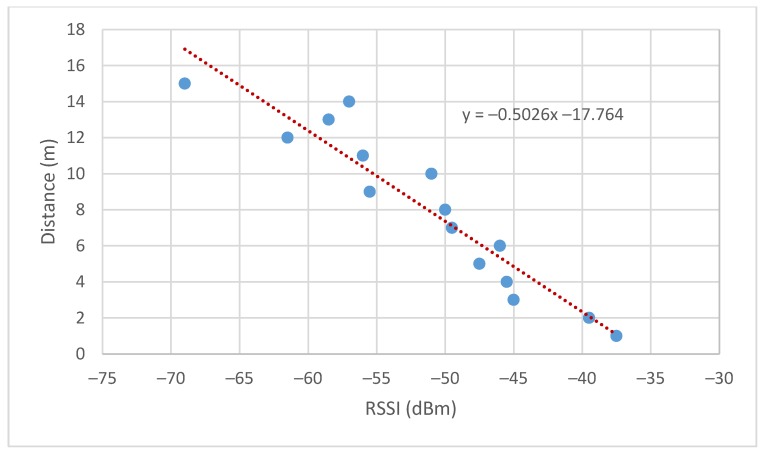
RSSI vs. distance node 2.

**Figure 15 sensors-19-03931-f015:**
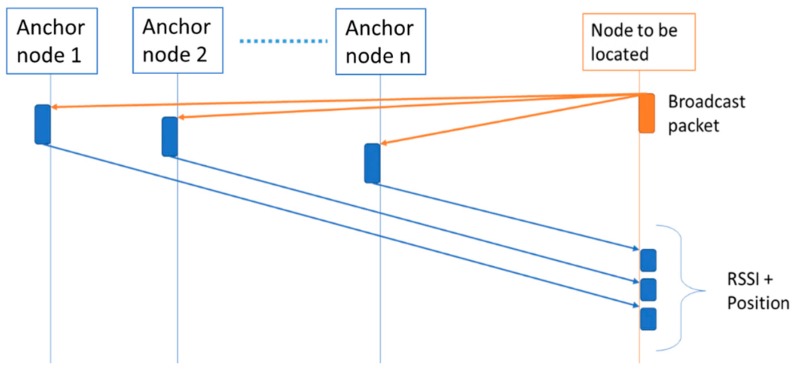
Communication between nodes.

**Figure 16 sensors-19-03931-f016:**
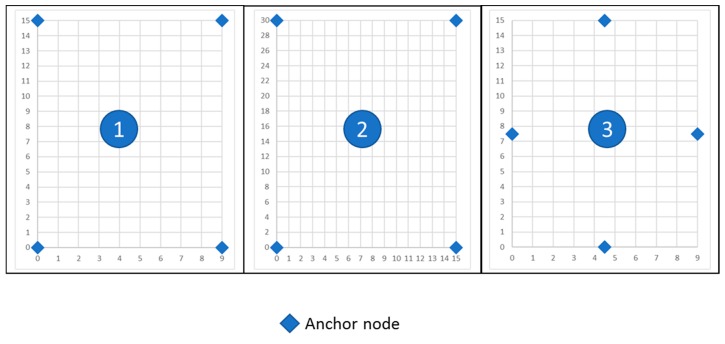
Experimental scenarios.

**Figure 17 sensors-19-03931-f017:**
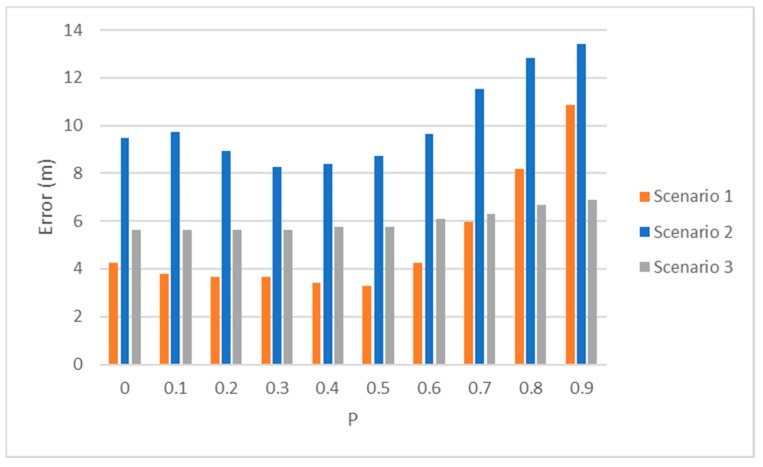
Localization error for different *p*-values for the three scenarios.

**Figure 18 sensors-19-03931-f018:**
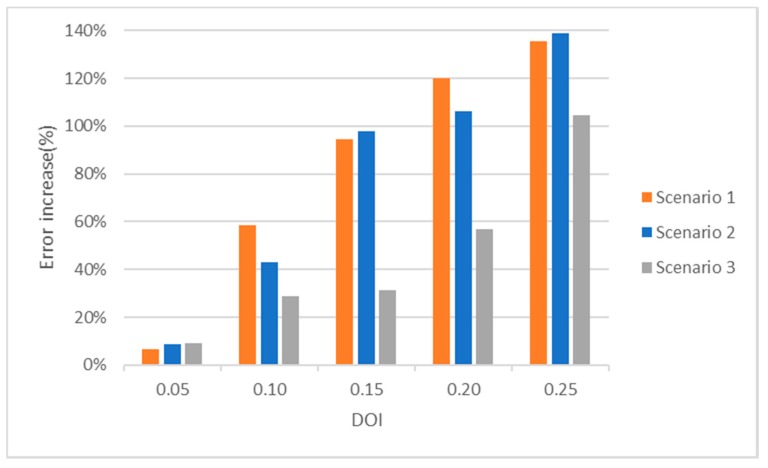
Error increase vs. DOI.

**Figure 19 sensors-19-03931-f019:**
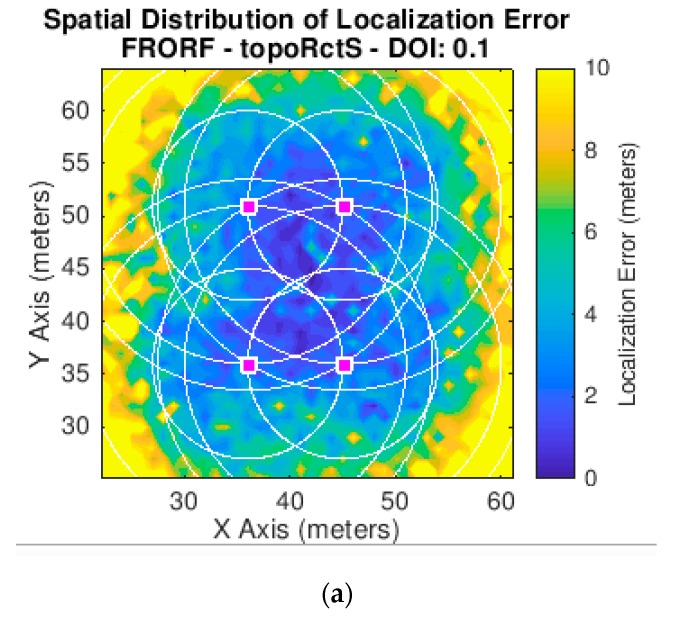
Experiments mean error results for scenario 1. (**a**) Scenario 1 in the simulator *(p*-value = 0.5); (**b**) scenario 1 in the outdoor set-up.

**Figure 20 sensors-19-03931-f020:**
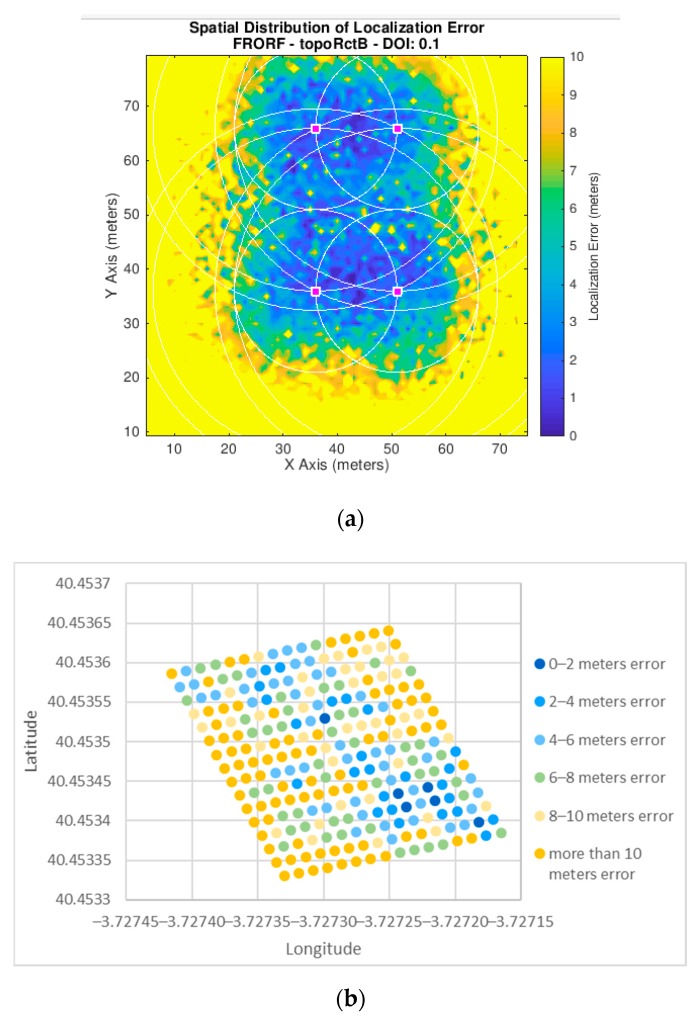
Experiments mean error results for scenario 2. (**a**) Scenario 2 in the simulator (*p*-value = 0.5); (**b**) scenario 2 in the outdoor set-up (*p*-value = 0.5).

**Figure 21 sensors-19-03931-f021:**
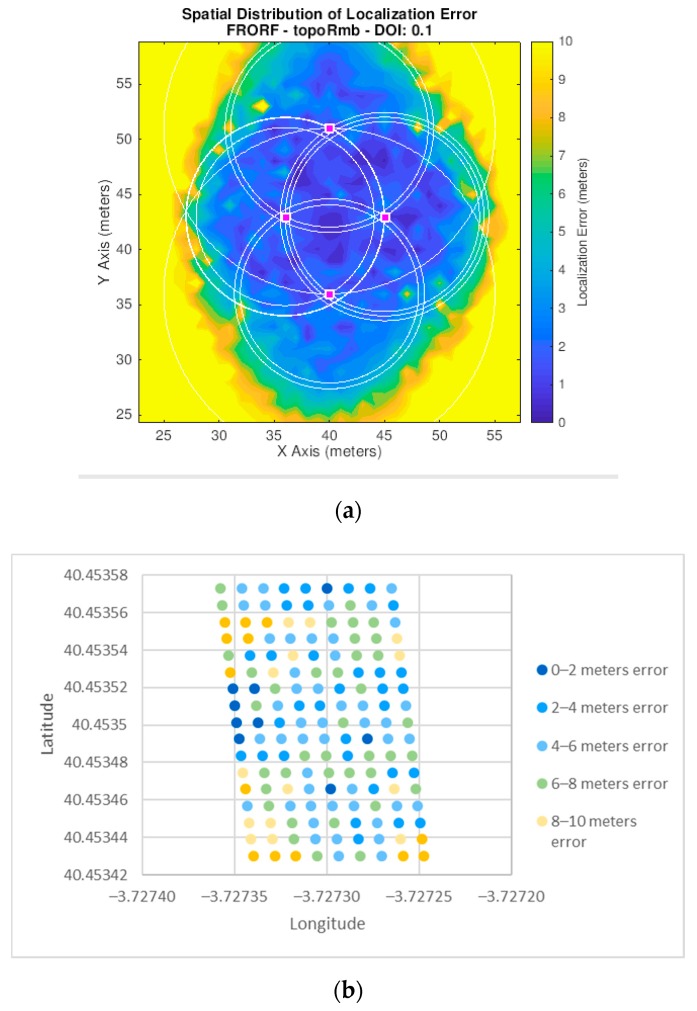
Experiments mean error results for scenario 3. (**a**) Scenario 3 in the simulator (*p*-value = 0.5); (**b**) scenario 3 in the outdoor set-up (*p*-value = 0.5).

**Figure 22 sensors-19-03931-f022:**
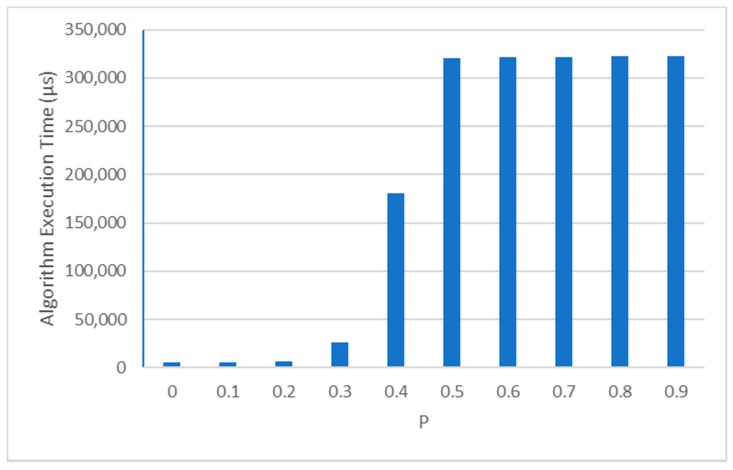
Execution time of fuzzy ring-overlapping range-free (FRORF) algorithm running in the YetiMote node for different *p*-values.

**Table 1 sensors-19-03931-t001:** Scenario conditions.

Scenario	Distance between Positions (*x*-axis)	Distance between Positions (*y*-axis)	Number of Measures per Position
1	1 m	1 m	10
2	1 m	2 m	100
3	1 m	1 m	100

**Table 2 sensors-19-03931-t002:** Average error.

Scenario	Simulation with Squares (DOI = 0.1)	Simulation with Circles (DOI = 0.1)	Simulation with Circles (DOI = 0.2)	Simulation with Circles (DOI = 1)	Real Set-Up
1	2.601 m	2.535 m	3.077 m	4.857 m	3.28 m
2	5.182 m	4.909 m	6.736 m	8.302 m	8.73 m
3	2.671 m	2.515 m	3.175 m	4.088 m	5.76 m

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
