# Peer review of "Experimental Evaluation of an RSSI-Based Localization Algorithm on IoT End-Devices"

_sensors, 2019, doi:10.3390/s19183931_

Round 1

Reviewer 1 Report

The paper presents the results of an experimental evaluation of an RSSI-based localization algorithm. The localization algorithm used is a modified and simplified version of the FRORF algorithm, adapted to implementation on resource-constrained devices. With FRORF the localization space is divided into a set of regions, and the device maps its position into a region by comparing its RSSI readings with those measured by anchor nodes.

The language in the paper and the paper organization are fair. The experimental results provided could be interesting to researches in the field of radio-based localization. However, the proposed algorithm modifications should be defined more formally and described more precisely. In particular:

(lines 201-202) “In order to simulate the FRORF algorithm, we use the grid scan method [23], which is a significant modification.”

In fact, FRORF already uses the grid scan method to derive the regional map (in the same way as presented in this paper), although it does not explicitly deal with the grid resolution. In this paper, the authors pointed out that the grid granularity impacts both algorithm precision and computation cost – this single statement cannot be considered as a significant modification.

The second modification relates to replacing rings with squares. I suppose that the square-based method replaces the grid scan method when the algorithm is implemented on resource-constrained devices, but it is not explicitly stated in the paper. The authors should precisely state the purpose of this algorithm, and what are its inputs and outputs. It is intuitively clear that the squares allow calculating CoGs directly instead to scan the whole localization space, but it is not immediately clear how the algorithm actually works. Therefore, a more detailed explanation of the algorithm operation would be beneficial.

The statement:

(lines 272-274:) “Therefore, for the calculation of the centroid of a region, all the CoG's of the rectangles that form it are calculated and the arithmetical average of these is made.”   

Is not true, in the sense that the arithmetic mean of CoGs of all rectangles that form a region does not always coincidence with the CoG of the region, especially if rectangles differ in size. The calculated CoG could only be an approximation of the real CoG.

Because the proposed modification represents an approximation of the grid scan algorithm, it would be interesting to see how this simplification impacts the localization accuracy. Unfortunately, such an analysis is not provided later in the Result section.

One of the main points of the paper is that the involved modification makes FRORF operation feasible on resource-constrained devices. However, in subsection 4.2.2 it is not quite clear what parts of the algorithm are executed by the devices. Does the device recreate the regional map after each package exchange, or does the device only create the map once? Is there a real need that the map is created by the device, or it can be created externally and downloaded into the device memory in an off-line phase? The authors should explicitly state what is included in the execution time.

In conclusion, the paper presents an attempt to adapt a fairly complex localization algorithm (FRORF) on resource-constrained devices, by indicating possible ways in which this can be done. The weak point of the paper is the presentation style. The involved modification should be described more clearly, formally and precisely.

Reviewer 2 Report

Please check minor typos (line 141: shows--> show)

Line 457: explain how to measure the distances in simulations

Reviewer 3 Report

In this manuscript, the author implemented and evaluated a FRORF-based localization algorithm in a real outdoor deployment to make its operation feasible on resource-constrained devices. The results show that this algorithm fit IoT end devices constraints, with low memory and computing capabilities.

       Strong points:

The author has detailed and clear descriptions of the FRORF algorithm and its implementation. The author introduces the related work in detail. And an extended literature survey has been made, which is very impressive. The language of the manuscript is clear and the description of experimental results is objective.

Weak points:

In section 5, how to get the conclusion that "p is close to 0.5 is optimal", please prove this conclusion with objective evaluation criteria; In section 5, what is the reason for setting the DOI parameter to 0.1? Please analyze the cause of the simulation results appropriately; There are some problems with the format of the manuscript: Please update the clarity of Figure 8; Please unify the size of the figures and formulas in the manuscript; The format of the manuscript needs to be uniform, such as line 371, line 389 and line 387. In recent years, there are paper “X. Cai, P. Wang, L. Du, Z. Cui, W. Zhang, and J. Chen, "Multi-objective 3-Dimensional DV-Hop Localization Algorithm with NSGA-II," IEEE Sensors Journal, 2019, DOI:10.1109/JSEN.2019.2927733.” improves DV-Hop localization. And it should be cited in this paper.

Round 2

Reviewer 1 Report

The authors have satisfactorily responded to all my questions and made the necessary changes to the paper. In particular, the rectangle-based algorithm for CoGs estimation is now explained more clearly and with sufficient details to allow others to replicate the proposed method. Also, the overall localization procedure is described more precisely. In my opinion, with these changes, the paper is of sufficient quality to merit publication in Sensors.

I do not have further comments.